# Withdrawing Antipsychotics for Challenging Behaviours in Adults with Intellectual Disabilities: Experiences and Views of Experts by Experience

**DOI:** 10.3390/ijerph192315637

**Published:** 2022-11-24

**Authors:** Gerda de Kuijper, Joke de Haan, Shoumitro Deb, Rohit Shankar

**Affiliations:** 1GGZ-Drenthe/Centre for ID and Mental Health, 9404 LL Assen, The Netherlands; 2Academic Collaboration ID and Mental Health, University Medical Centre Groningen, 9713 GZ Groningen, The Netherlands; 3Faculty of Medicine, Department of Brain Sciences, Imperial College London, London SW7 2BX, UK; 4Peninsula School of Medicine, University of Plymouth, Plymouth PL4 8AA, UK

**Keywords:** intellectual disabilities, antipsychotics, challenging behaviour, discontinuation, experts by experience views, qualitative study, interviews

## Abstract

People with intellectual disabilities (PwID) are frequently prescribed long-term antipsychotics for behaviours that challenge (BtC) despite the lack of proven effectiveness and the increased risks for side effects of these medications in this population. National and international good clinical practice guidelines recommend deprescribing antipsychotics for BtC, which is often not successful due to environmental and other factors. The involvement of all stakeholders, including PwID, is crucial for deprescribing. However, studies showed that PwID and/or their families are often not involved in decision-making regarding the (de)prescribing of antipsychotics despite their desire to get involved. Moreover, studies on the views of PwID regarding their experiences of withdrawing from antipsychotics are lacking. The aim of this study was to gain insight into the views of PwID by investigating their experiences of discontinuation of long-term prescribed antipsychotics for BtC. A qualitative study was set up. Seven experts by experience with mild intellectual disabilities were interviewed. After six interviews, data saturation was achieved. Interviews were transcribed verbatim. Using phenomenological analysis, themes on lived experiences were extracted. Each consecutive interview was analysed. The four main themes extracted from the interviews were the quality of treatment, knowledge and information about psychotropics and the process of withdrawal, support from the participants’ environment and the coping style of the interviewees themselves.

## 1. Introduction

People with intellectual disabilities (PwID) frequently engage in behaviours that challenge (BtC), especially those with higher care needs and/or who receive residential care [1]. BtC include aggressive behaviour (e.g., verbal abuse, threats and physical violence), destructive behaviour (e.g., breaking or destroying properties and other objects and setting fires), disruptive behaviour (e.g., repetitive screaming, smearing faeces, setting off fire alarms when there is no fire, calling the emergency services when there is no emergency), self-injurious behaviour (e.g., self-biting, head-banging) and sexually harmful behaviour (e.g., sexual assaults, rape and stalking), as described in the National Institute of Care Excellence (NICE, NG11) [2] guidelines.

There are many reasons for BtC, including biological (genetic disorders, physical problems including pain, medication side effects), psychological (psychiatric disorders including anxiety, depression, psychosis and post-traumatic stress disorders) and social factors (inappropriate accommodation and lack of support) [3]. The reported prevalence of psychiatric disorders in PwID varies between 14.4 and 60% depending on the definition of psychiatric diagnosis used (e.g., the inclusion or exclusion of BtC, ADHD and ASD within the psychiatric disorder diagnosis), method of detection using different classification systems (e.g., ICD vs. DSM), use of assessment methods (screening instrument vs. structured interviews), population studied (institutionalised vs. clinic-based vs. community-based), etc. [4,5,6]. Neurodevelopmental disorders, such as autism spectrum disorder (ASD) and attention deficit hyperactivity disorder (ADHD), are common among PwID [7]. The rate of both psychiatric disorders and BtC increases in PwID with these comorbidities [7,8]. However, both false positive and false negative diagnoses of psychiatric disorders are common in PwID [6].

In the UK, NHS England (NHSE) embarked on a major campaign six years ago called ‘STopping Over-Medication of People with learning disabilities, autism or both (STOMP)’ [9] to which STAMP (supporting treatment and appropriate medication in paediatrics) was also added recently. The Royal College of Psychiatrists in the UK also published a position paper to support the STOMP STAMP initiative [10]. The National Institute for Health and Care Excellence (NICE) in the UK [2] and the World Psychiatric Association [11] developed guidelines for the use of psychotropics to address BtC among PwID, including recommendations for the initiation, monitoring and potential withdrawal of psychotropic medications.

BtC is best-treated non-pharmacologically, but in some situations, short-term prescriptions of psychotropics may be beneficial [2]. However, BtC is often treated with antipsychotics and prescribed long-term in the absence of a valid psychiatric diagnosis [12,13] despite the lack of evidence for its long-term effectiveness for BtC [14,15,16,17]. Moreover, PwID are vulnerable to the side effects of antipsychotics, especially neurological side effects, such as movement disorders; sedation/sleepiness; and metabolic side effects, such as obesity and diabetes [15,18,19,20]. Therefore, it is important to balance the pros and cons of antipsychotic use carefully and to discontinue them when there is no longer a valid indication, especially in the case of BtC [2].

Furthermore, it is important to involve all the stakeholders. This means that along with staff and healthcare professionals, the PwID themself, their family members and other representatives of PwID showing with BtC should get involved in the decision-making of the (de)prescribing of psychotropic drugs [21]. Indeed, studies showed that PwID and their family members would like to contribute to treatment decisions about BtC and medication use but they are often not involved in decision-making regarding the (ongoing) prescription of medication [22,23,24].

Although national and international guidelines recommend [2,11,25] that long-term antipsychotics for BtC should be deprescribed, discontinuation often fails [26,27,28,29] because of a variety of factors. It is important to involve all stakeholders to increase the proportion of successful discontinuation, as was shown in a study by Shankar and colleagues [28]. In particular, the role of the primary caregivers and other staff should be recognised as the key factors that influence the successful discontinuation of antipsychotics [30,31,32], as staff may lack knowledge of the effects and side effects of antipsychotics and may have unrealistic expectations of the beneficial effects of psychotropic medications on the psychological functioning of their clients [30,31]. In a recent study of interviews with support staff in the UK, some direct care staff felt that medication is necessary to control BtC in PwID, whereas others felt that these medications were ‘chemical restraints’ [32].

Furthermore, evaluation of the process of discontinuation is important to identify potential barriers and facilitators for the deprescribing of these agents. A survey among UK psychiatrists showed that family and staff concerns, lack of multidisciplinary input and unavailability of psychosocial interventions were barriers to attempting to discontinue antipsychotics [33]. A Dutch study among direct support professionals about their perceptions of the effect of antipsychotics on BtC and their willingness to cooperate in discontinuing these agents showed that they believed that antipsychotics were effective in the management of BtC [34]. However, they also observed the negative consequences of side effects. Staff were willing to cooperate in reducing the antipsychotic drug use but were not confident about successful complete discontinuation [34].

The current study aimed to add knowledge about facilitators and barriers to the successful discontinuation of antipsychotic drug use for BtC in PwID. Although studies were done on the experiences and views of prescribers and support professionals regarding withdrawing antipsychotics for BtC, as far as we know, studies on users’ perspectives are missing. Therefore, in this study, we investigated the views of people with mild ID on the discontinuation of their antipsychotic drugs and their experiences with the withdrawal process.

## 2. Methods

### 2.1. Design and Participants

This was a qualitative interview study. Potential participants were selected by their clinicians according to preset eligibility criteria. Eligible participants were adults aged between 18 and 65 years and had mild intellectual disabilities according to the DSM5 classification system (Diagnostic and Statistical Manual of Mental Disorders, 5th revised edition; American Psychiatric Association, 2013). They had to be currently undergoing withdrawal of antipsychotic medications that they had been receiving for more than six months, where either their antipsychotics had already been discontinued recently or they had gone through an attempted withdrawal in the recent past. Potential participants were required to have competency and language skills to take part in the interviews and be able to express their views in the Dutch language. Participants with dementia, bipolar disorder, schizophrenia and chronic psychosis were excluded.

### 2.2. Procedures

Potential participants were recruited from the responders of a survey that had previously been carried out among Dutch physicians specialising in intellectual disabilities, psychiatrists and mental healthcare specialist nurses [35]. Additionally, the first two authors recruited participants from their networks and by asking their colleagues at the outpatient clinic of the mental healthcare organisations they were affiliated with that provide services for people with intellectual disabilities.

Potential participants were contacted on the phone or visited by the second author (J.d.H.). During these calls or visits, they were informed about the goal and content of the interviews. In these meetings and the interview sessions, the interviewer (J.d.H.) used accessible language, such as short sentences and clear questions. Furthermore, the interviewees could ask a significant other, such as their personal support professional or family member, to accompany them during the interviews.

A topic guide was used for these interviews (see Table 1), the items of which were drawn from the relevant literature on the subject [21,22,23,24,25,36,37].

A test interview was done by the second author (J.d.H.) to assess the accessibility of the content and comprehensibility of the interview questions. This test interview took place in the inpatient clinic of the mental healthcare organisation the interviewer was affiliated with. The interviewee was a client who was currently undergoing withdrawal from her antipsychotics. This interview was not recorded and no data from this interview were included in the results section. The test interviewee revealed that there was no need to adapt the topic guide, language use or other subjects.

### 2.3. Ethics

All participants consented to take part in the interviews and signed an informed consent form before the interview took place. They were informed that the encrypted data would be stored at a safeguarded place and analysed and published anonymously and that they could withdraw from the study at any moment without giving a reason.

The interview study was not a medical, scientific study according to the Dutch Act on Medical Studies with Human Beings, as no personal health-related data were collected, but only the participants’ views were captured through these interviews. Therefore, no ethical approval was required for this study. The study was carried out under the responsibility of Mental HealthCare Drenthe (GGZ Drenthe) and was approved by their research committee (declaration is available on request). All data were collected, stored and safeguarded anonymously according to the European Act on Protection of Personal Information, which, in the Netherlands, was ratified in 2018.

### 2.4. Analyses

The interviews were video-recorded and transcribed verbatim. A phenomenological data analysis method was used to extract themes from the transcriptions by means of (a) thorough reading and re-reading of the transcripts to obtain a general sense of the whole content; (b) extraction of significant statements on the phenomenon under study and formulation and coding of the meaning of these statements; (c) sorting of categories, clusters of themes, and themes of these codes and meanings; and (d) integration of the findings and description of the fundamental structure of the phenomenon that ought to be validated by the interviewees [38].

The sample size was established according to the criteria of data saturation in qualitative studies, which means that qualitative data were collected to the point where a sense of closure was attained because new data yielded redundant information [39]. In the present study, each consecutive interview was analysed. When the last interview yielded no new codes and themes, data saturation was achieved and no new interviews/new participants were needed.

Transferability was maintained (via a thorough reading and re-reading, as well as a rich description of the content of the interviews), along with credibility; confirmability; authenticity via peer-review of the coding of the meanings, categories and themes of the statements; and dependability via the interviewer’s log-keeping [40].

Data were analysed by the second author (J.d.H.). A random sample of three interviews (out of seven in total) was independently analysed by the first author (G.d.K.) to assess the agreement with the analyses of J.d.H.

## 3. Results

Seven experts by experience with mild intellectual disabilities participated in the interview study. These were three men and four women, whose ages ranged from 22 to 62 years. Three used quetiapine; two used aripiprazole; one used olanzapine; and one used a combination of haloperidol, quetiapine and clozapine. All participants were white. The duration of the interviews was 25 min on average. After each interview, the extraction of significant statements and coding and categorising their meanings took place. After the sixth interview, no new categories and themes were extracted; therefore, data saturation was achieved. Four main themes could be extracted from the content of the interviews. These were (a) the quality of the treatment, (b) knowledge and information about the process of withdrawal, (c) the conditions that needed to be met before a discontinuation process could start and (d) the coping style of the person whose medication was discontinued.

(a)The quality of treatment.

All participants indicated that they had used antipsychotics for many years because of various reasons, but none was for a licensed indication. For example, one participant stated:


*‘Finally, after many years, it was found that I have ADHD, and then quetiapine won’t ’work very well, and Concerta will work better, you might say.’*


The reasons to discontinue antipsychotics were mainly because participants judged that they no longer had symptoms that needed treatment and/or because of side effects, such as weight gain, tremors, flattened affect and sleepiness. Participants stressed the importance of receiving good treatments from clinicians who take them seriously and provide a clear and understandable explanation about their treatment and the reason for the prescription of their medication, such as (i) What is it for? (ii) How does it work? (iii) What side effects could occur? (iv) When to consult my doctor? However, often they had not been provided explanation and information when the medicine was newly prescribed.

(b)Knowledge and information about the process of withdrawal.

Most participants had some fear of discontinuing because they were afraid that their behavioural problems may re-emerge.


*‘I was afraid it turned out to go wrong.’*


All participants indicated that they and their caregivers were provided with oral information by their doctor or nurse about the discontinuation schedule and dose reductions and the possibility of withdrawal symptoms and changes in behaviour. Four participants experienced withdrawal symptoms, and most of them knew this could happen and what kind of symptoms could be expected.


*‘I was like, this is just part of it, and then I went reading or doing something else and then it disappeared.’*


All participants had informed their caregivers, peers or colleagues that they were discontinuing their drugs and the possibility of withdrawal symptoms.


*‘They have to know I may react differently.’*


(c)Conditions to be met before starting a discontinuation process according to participants’ experiences.

Half of the participants found the withdrawal process difficult for various reasons. One participant had problems with providing care for her children, others had physical and emotional problems. In one case, a dose increase of the antipsychotic drug was necessary to relieve these symptoms.

All participants indicated the importance of a good, accessible and trusted relationship with their doctors. This was helpful in continuing their withdrawal process when they had questions or wanted to talk about negative feelings when they experienced difficulties with the withdrawal process.


*‘All went well in agreement, I could always phone or come for consultation’.*


Moreover, others in the participants’ environments were important for providing support.


*’Support from my parents, brother, sister-in-law, grandmother and grandfather.’*



*‘Good to talk about it with others around you, anyway I think this is important because you won’t make it on your own’.*


(d)Coping style of the participant.

Most participants indicated that an appropriate mindset and intrinsic motivation are important to start with and continue the discontinuation process.


*‘Yes, you have to switch and stick to it.’*



*‘Just say to yourself, I can make it.’*



*‘It’s also your own mindset, how to deal with yourself when you are afraid to discontinue, then it will be difficult. This was my experience, when you switch your mindset appropriately, then it will be all right.’*


Participants also provided tips and advice for their peers. ‘Ask help’.


*‘When you are angry, go to your support professional’, ‘stick to a structure in your daily activities.’*


## 4. Discussion

In this study, we explored the experiences and views of adults with mild ID regarding the withdrawal from their long-term-used antipsychotic drugs prescribed for BtC. This process may be difficult because PwID are often dependent on others and/or easily influenced by others and often need support in decision-making around deprescribing and maintaining the process of discontinuation [22,24,41].

Recent studies in the UK and the Netherlands have shown that even after long-term use, it is possible to discontinue antipsychotic medication in 25–61% of adults with intellectual disabilities and have a 50% dose reduction in another 11–19%, although, in up to 20% of cases, antipsychotics were re-instated within 3–4 years of discontinuation, primarily due to the resurgence of BtC [15,26,28]. However, several internal (e.g., comorbid ASD, baseline severity of BtC, an underlying psychiatric disorder) and external (e.g., baseline high dose of antipsychotics, lack of support and contingency plan during the withdrawal process) factors determine the success of any antipsychotic withdrawal [27,42].

The resurgence of BtC may not be related to the withdrawal of medication per se. The withdrawal may unmask a previously undiagnosed psychiatric disorder, such as psychosis, depression or bipolar disorder, or lead to a relapse of them. Discontinuation or dose reduction may lead to withdrawal syndromes, such as insomnia, anxiety and panic. A supersensitivity syndrome consisting of extrapyramidal symptoms, such as akathisia, Parkinsonism and dyskinesia, was associated with antipsychotic withdrawal [43,44]. All these can lead to BtC. In some cases, caregivers’ anxiety may exacerbate the BtC or its perception, leading to the enhanced reporting of BtC. Therefore, the clinician will need to make a full assessment of the causes and effects of BtC using standard methodologies [45] (https://spectrom.wixsite.com/project/ (accessed on 1 November 2022).

It is known from other studies that the involvement of all stakeholders in decisions around the withdrawal of long-term-used antipsychotics for BtC is a key factor for success [22,23,24,25]. However, the most important factor may be the involvement of family caregivers and patients with ID themselves. Studies have shown that even though family caregivers have the appropriate knowledge of the BtC of their relatives and they would like to participate, they are often not involved in treatment decisions, e.g., psychotropic drug prescribing [21,41]. Furthermore, individuals with ID themselves often lack information and/or are not asked to contribute to decisions in the prescription of antipsychotics [22,23,24,25,41]. Although previous studies of interviews of adults with ID have touched on the issue of psychotropic medication use [41], as far as we know, the current study is the first one that specifically targeted the experience of PwID who have gone through the experience of psychotropic withdrawal.

In the current study, a phenomenological analysis of interviews with seven experts by experience extracted four themes that may be of importance in the successful discontinuation of long-term antipsychotic drugs. These themes were (a) the quality of treatment, (b) knowledge and information about the discontinuation process, (c) the conditions that should be met in a withdrawal process according to participants’ experiences of taking part in the withdrawal process and (d) the participants’ own coping style.

The interviewed participants indicated that a good relationship with their doctor was important. This meant that they were taken seriously and had been provided information about the indications, effects and side-effects of the antipsychotic agents they received, the reason why this medication was no longer indicated/could be discontinued and information about the discontinuation process. Furthermore, their doctor was reliable, had a personal approach and was accessible. The prescribing doctor had to be available when there were problems during the discontinuation process, e.g., withdrawal symptoms or negative feelings. Furthermore, interviewees mentioned that support from their peers, family and support professionals was important. The finding of the fourth theme, namely, the interviewees’ own coping style as an important factor in successful discontinuation, was new and surprising. This may suggest that promoting and stimulating patients’ feelings of self-confidence and self-efficacy in the management of and decisions around their own medication use may be helpful in the successful discontinuation of inappropriate (long-term) medication. Perhaps, there is a role for support professionals and healthcare workers to support PwID to achieve this. A recently developed training programme for support staff to help with the deprescribing of psychotropics for people with ID, which is called SPECTROM (https://spectrom.wixsite.com/project/ (accessed on 1 November 2022), developed resources that could be used for this purpose [30].

Our study had some limitations. For the interview study, we had to recruit participants from our own networks, which may have influenced the representativeness of participants in the study. For example, all participants were white. Moreover, we did not interview representatives of people with moderate, severe and profound intellectual disabilities who discontinued antipsychotics; therefore, the results of this interview study are not applicable to these populations. Moreover, we only had seven participants in this qualitative interview study. However, we may assume that this sample size was sufficient because data saturation was achieved in the sixth interview. Finally, because this study was carried out among Dutch clients in Dutch healthcare settings, the results might not be generalisable to other countries. Yet, it may be assumed that the occurrence of withdrawal symptoms and the need for support during the process of medication withdrawal are universal among all people with mild intellectual disabilities. Therefore, the results of this study may also be applicable to other countries with comparable service- and care-providing systems for PwID.

## 5. Conclusions

The experiences and opinions of experts by experience with mild intellectual disabilities who had discontinued their antipsychotic drugs were that the reliability and accessibility of their doctors and being taken seriously; a clear explanation and information about the discontinuation trajectory; support from peers, families and care professionals; and their own coping styles were important factors in successful discontinuation.

In addition to the education of care professionals in supporting their clients with mild intellectual disabilities in the decision-making around their medication use, the clients’ own coping style in decisions around the start of and management of difficulties during an antipsychotic discontinuation process may be an important factor for the successful completion of discontinuation.

## Figures and Tables

**Table 1 ijerph-19-15637-t001:** Topic guide with subjects for interviews on users’ experiences with the withdrawal of antipsychotics.

Topic Guide
Reasons for antipsychotic drug use
Reason for discontinuation of antipsychotics
Information the participants received about their medication use and the discontinuation process
Mental and physical health problems experienced during the discontinuation process
Opinions of the participants about the decision and process of discontinuation
Tips or advice from the participants for the professionals and other PwID about the discontinuation process

## Data Availability

The data are available on request and with a substantiated explanation at the GGZ Drenthe department research website (https://ggzdrenthe.nl/research, accessed on 1 November 2022).

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
