# Peer review of "Withdrawing Antipsychotics for Challenging Behaviours in Adults with Intellectual Disabilities: Experiences and Views of Experts by Experience"

_ijerph, 2022, doi:10.3390/ijerph192315637_

Round 1
Reviewer 1 Report
Thank you very much for giving me the opportunity to review your manuscript. Withdrawal of antipsychotics in people with intellectual disabilities and challenging behaviour is a regular discussion´s topic, which raises some doubts in the daily practice. On the other hand, take into account the subjective opinion of the participants about this question is something, is something that I consider really important to improve our daily work.
In spite if that, I have some recommendations, which I consider could help to improve the quality of the manuscript.
Introduction:
I miss some references in the introduction about underlying psychiatric disorders, which could play a role in challenging behaviour in people with intellectual disabilities (for instance, affective or psychotic disorders). I find also a lack of information about autism spectrum disorders and challenging behaviour. It would be not the same withdraw psychotropic drugs in a pacient with ASD, ID and challenging behaviour or with a comorbid psychotic disorder, than in somebody without. Some useful references could be: Peña-Salazar et al. (2020), Peña-Salazar et al. (2022), Mc Carthy et al (2010), Mazza et al. (2020).
Methods: I would include the psychiatric diagnosis of the participants in this study.
Discussion: I consider, the discussion could be enriched with the inclusion of some references about challenging behaviour and comorbid psychitric disorders.
Thank you very much,。
Reviewer 2 Report
I think this is a well-written paper, of importance - given initiatives to redue antipsychotic prescribing - and of interest to the readers. I only have minor comments on how it could be improved.
- I think that reference to the STOMP initiative would help to set the scene for the move towards describing antipsychotics.
- The gender/age distribution looks sensible but I suspect that all participants were white. I think this should be specified as a limitation.
- Six interviews seems low (to me) to reach saturation. Are the authors confident that this is sufficient?
- (Very minor) There is a closing " ' " missing on line 47.
